# Sequences of purchases in credit card data reveal lifestyles in urban populations

Riccardo Di Clemente [1,2], Miguel Luengo-Oroz[3], Matias Travizano[4], Sharon Xu[1], Bapu Vaitla[5] & Marta C. González[1,6,7]

Zipf-like distributions characterize a wide set of phenomena in physics, biology, economics, and social sciences. In human activities, Zipf's law describes, for example, the frequency of appearance of words in a text or the purchase types in shopping patterns. In the latter, the uneven distribution of transaction types is bound with the temporal sequences of purchases of individual choices. In this work, we define a framework using a text compression technique on the sequences of credit card purchases to detect ubiquitous patterns of collective behavior. Clustering the consumers by their similarity in purchase sequences, we detect five consumer groups. Remarkably, post checking, individuals in each group are also similar in their age, total expenditure, gender, and the diversity of their social and mobility networks extracted from their mobile phone records. By properly deconstructing transaction data with Zipf-like distributions, this method uncovers sets of significant sequences that reveal insights on collective human behavior.

[1] Department of Civil and Environmental Engineering, Massachusetts Institute of Technology, Cambridge, MA 02139, USA. [2] The Bartlett Centre for Advanced Spatial Analysis, University College London, London WC1E 6BT, UK. [3] United Nations Global Pulse, 46th Street and 1st Avenue, New York, NY 10017, USA. [4] GranData, 550 15th Street Suite 36C, San Francisco, CA 94103, USA. [5] Department of Environmental Health, Harvard University, 677 Huntington Avenue, Boston, MA 02115, USA. [6] Department of City and Regional Planning, Berkeley, CA 94720-1820, USA. [7] Lawrence Berkeley National Laboratory, 1 Cyclotron Road, Berkeley, CA 94720-1820, USA. Correspondence and requests for materials should be addressed to M.C.Gál. (email: martag@mit.edu)

In the age of information, we leave digital traces of our everyday activities: the people we call, the places we visit, the things we eat, and the products we buy. Each of these human activities generates data that when analyzed over long periods yield a comprehensive portrait of human behavior[1–6].

In the past decade, call detailed records (CDRs) have been of paramount importance to understand the daily rhythms of human mobility[7–11]. By properly analyzing billions of digital traces, our modern society has a whole framework to analyze wealth[12], socio-demographic characteristics[13], and to better tackle the origins of urban traffic[14,15]. By contrast, we still need to better exploit the credit card records (CCRs) to uncover the behavioral information they may hide. Main uses of CCRs have been to measure similarity in purchases via affinity algorithms[16,17]. Recent research has also shown that credit card data can be used analogously to mobile phone data to detect human mobility. Namely, the CCRs inform us about the preferred transitions between business categories, identifying the unevenness of the spatial distributions of people's most preferred shopping activities[18], and to enrich urban activity models. Consumers' habits are shown to be highly predictable[19], and groups that share work places have similar purchase behavior[20]. These results allowed defining the spatial–temporal features to improve the estimates of the individual's financial well-being[21].

It has been measured by individual surveys and confirmed by credit card and cash data that the vast majority of daily purchases is dominated by food and then followed by mobility and communication–social activities[13,22]. Their frequency seems to follow Zipf distribution, meaning that the most frequent category of purchases will occur approximately twice as often as the second most frequent category, three times as often as the third, etc. Grouping the consumers depending on their socio-demographic attributes preserves the Zipf-like behavior and dominant purchase (food). For each group, there is a peculiar order in the abundance of less frequent category. As pointed out by Lenormand et al.[13] and Sobolevsky et al.[23] this depends on the socio-demographic features such as income, gender, and age.

Hence, the challenge at hand is to obtain meaningful information within these highly uneven spending frequencies to capture a comprehensive picture of their shopping styles related to socio-economic dynamics within the city.

A similar challenge appears in the sequence of diseases in the medical records[24] or phenotype associations with diseases[25]. Existing approaches cluster patients based on their historical medical records described by the International Classification of Diseases. In this case, the frequency-inverse document frequency (TF-IDF) ranking is used to eliminate redundant information.

In the matter of uneven word frequency in the text corpora[26], Bayesian inference methods have been used to detect the hidden semantic structure. In particular, the latent Dirichlet allocation (LDA)[27] is a widely used method for the detection of topics (ensemble of words) from a collection of documents (corpus) that best represent the information in data sets.

However, both of the above-mentioned approaches do not take into account the temporal order in the occurrence of the elements. Our goal is to eliminate redundancy while detecting habits and keeping the temporal information of the elements, which in the case of purchases are an important signature of an individual's routine and connect them to their mobility needs. In this work, we identify significantly ordered sequences of transactions and group the users based on their similarity. This allows offering deeper description of consumer behavior, unraveling their routines.

In this work, we are interested in uncovering diverse patterns of collective behavior extracted from this data. Specifically, how the digital footprint of CCRs can be used to detect spending habits, reflecting interpretable lifestyles of the population at large. By integrating credit card data with demographic information and mobile phone records, we have a unique opportunity to tackle this question.

The presented method is able to deconstruct Zipf-like distribution into its constituent's distributions, separating behavioral groups. Paralleling motifs in network science[28], which represent significant subnetworks, the uncovered sets of significant sequences are extracted from the labeled data with Zipf-type distribution. Applied to CCRs, this framework captures the semantic of spending activities to unravel types of consumers. The resulting groups are further interpreted by coupling together their mobile phone data and their demographic information. Consistently, individuals within the five detected groups are also similar in age, gender, expenditure, and their mobility and social network diversity. We show that the selection of significant sequences is a critical step in the process; it improves the TF-IDF method that is not able to discern the spending habits within the data. Remarkably, our results are comparable with the ones obtained by LDA, with the added advantage that it takes into account the temporal sequence in the activities.

## Results

**Data analysis**. We analyze individual CCR transactions over 10 weeks in 150,000 users who live in one of the most populated cities in Latin America (Mexico City, Mexico). The data set contains age, gender, and residential zipcode of the users (Supplementary Figure 1A–C). For each user, we analyze the chronological sequence of their transactions and the associated expenditure labeled with the transaction type via a Merchant Category Code (MCC)[29]. The purchase entries are aggregated by the user and are temporally ordered with respect to each day. For one-tenth of the analyzed users, we also have their CDR data over a period of 6 months (overlapping the CCR time period), including time, duration, location of the calls, and identification of the receiver. While payment with cards and electronic payment terminals are being promoted in the region to improve financial inclusion, credit card adoption rates remain relatively low at 18% for the population[30]. First, we check how representative the CCR users are within the city. We observe the correlation between the median CCR expenditure in the data set at the district level and the average monthly wage in the same district, according to the census (Fig. 1a) (Source: INEGI, National Survey of Occupation and Employment (ENOE) and population aged 15 years and older.). The monthly expenditure of card users is high in relation to their monthly wages, indicating that the adoption of credit cards predominantly occurs among users with higher wages in each district. However, our users' sample spans over all the city districts with different income levels. We observe that wider adoptions of credit card are across male and young adults (aged 35–50 years) in each district (Supplementary Figure 1B–F). The spending patterns in the CCRs reveal that the frequency of the purchase types follows Zipf's law (Supplementary Figure 2A). The majority of shoppers use more frequently the top 20 transactions codes presented in Fig. 1b, among hundreds of possible MCCs. Moreover, slight variations emerge in this trend when dividing the population by wealth, age, and gender (Fig. 1c). In general, transaction codes related to food, mobility, and communication, in that order, dominate the number of top transactions in all groups and the number of transactions per day; for each user is not affected by any socio-demographic category (Supplementary Figure 2B, C).

**Credit card transaction codes as sequence of words**. Our main goal is to amplify the signal in the data to identify the individuals'

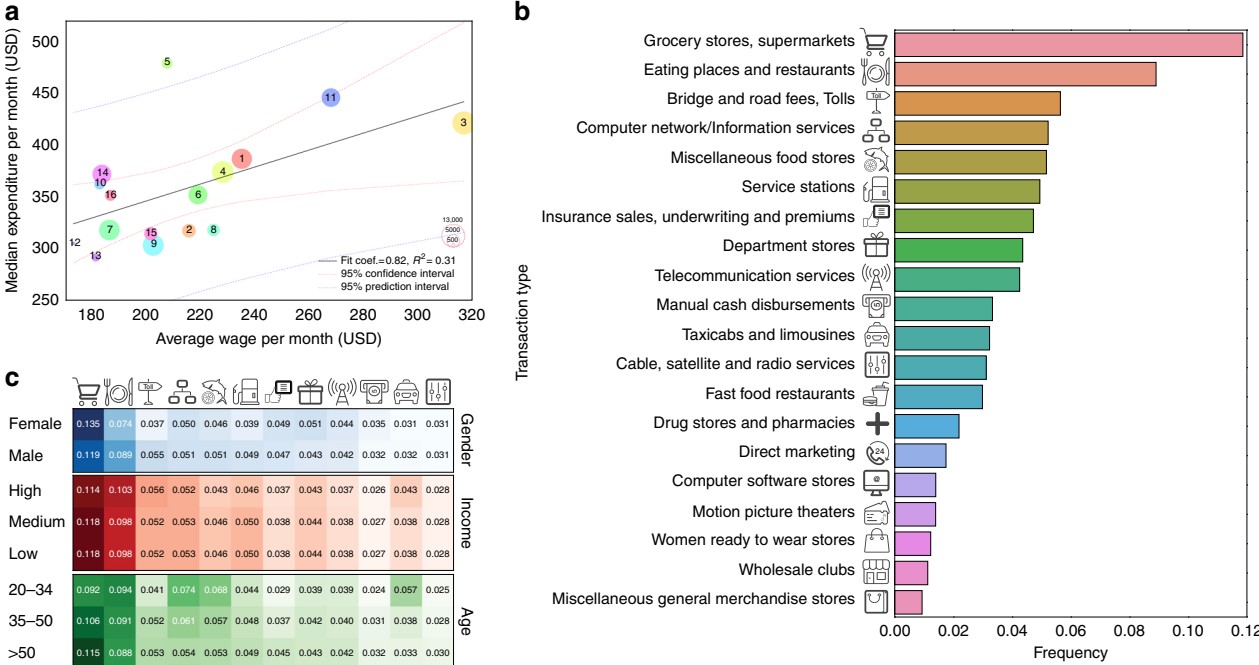

**Fig. 1** Transaction frequency by type and their demographics. **a** User median expenditure per month in CCR transactions vs. the average monthly wage in their district of residence. The color and the number represent different districts of Mexico City (see Supplementary Figure 1), and the size of the circles is proportional to the number of users in the district. **b** Transactions by type as defined by MCC[29]. **c** Comparison of frequencies by transaction types (same as in **b**) separating users in groups according to their gender, income, and age. The share of transaction frequency is distributed similarly among different groups. The icons used in this figure are work of Azaze11o/Shutterstock.com

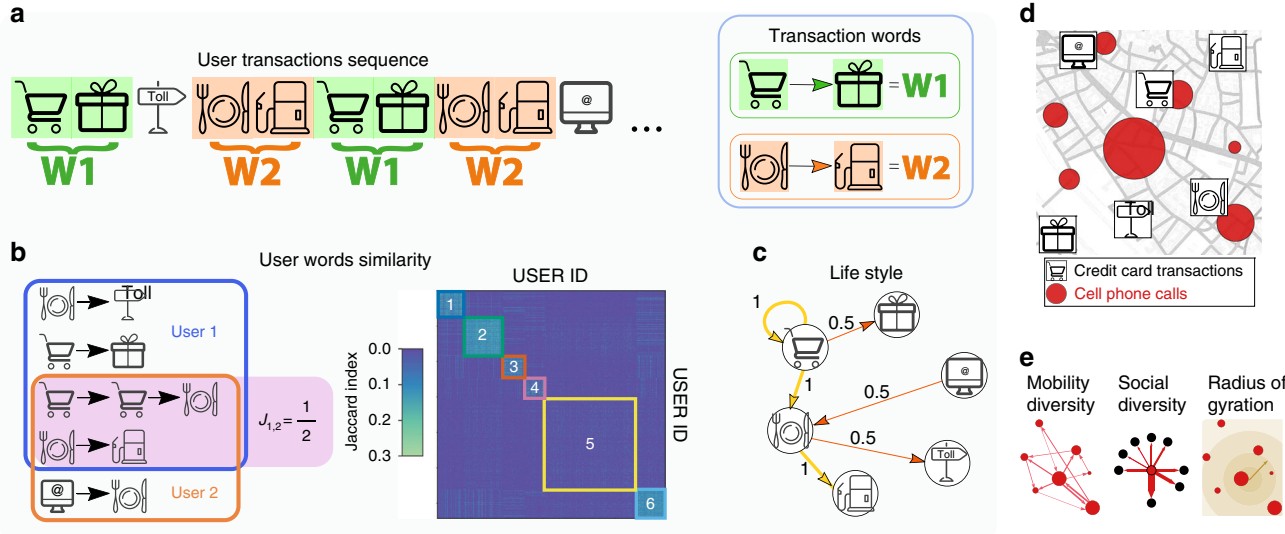

**Fig. 2** Methods and metrics. **a** Schematic representation of the Sequitur's algorithm applied to a sequence of transactions of one user to detect words and identify significant transaction sequences in the data set. **b** Calculation of the similarity between two users (left) based on the Jaccard index of their significant sequences to define the matrix of users' similarity (right). Group of users are detected based on similar sequences of transactions. **c** Lifestyle representation based on sample users 1 and 2 of **b**. **d** Example of traces of CDR and CCR data for the user. **e** Metrics adopted for the analysis of CDR data. The icons used in this figure are work of Azaze11o/Shutterstock.com

expenditure habits hidden in the non-uniform distribution of transaction types present in a Zipf's type of distribution. The first step in this direction is to transform the chronological sequence of user MCC codes into a sequence of symbols given by the transaction codes (Fig. 2a). We apply the Sequitur algorithm[31] to infer a grammatical rule that generate words, defined as MCC symbols that repeat in sequence. The result of this process applied

recursively is a compression of the original sequence with new symbols called words, which offer insights into the repeated sequences of transactions. We take each word as a routine in shopping, as they are a chronological sequence of two or more MCCs that appear frequently. We detect more than 10,000 different words also following a Zipf-type distribution, as presented in Fig. 3. We noticed that the inter-time transactions between

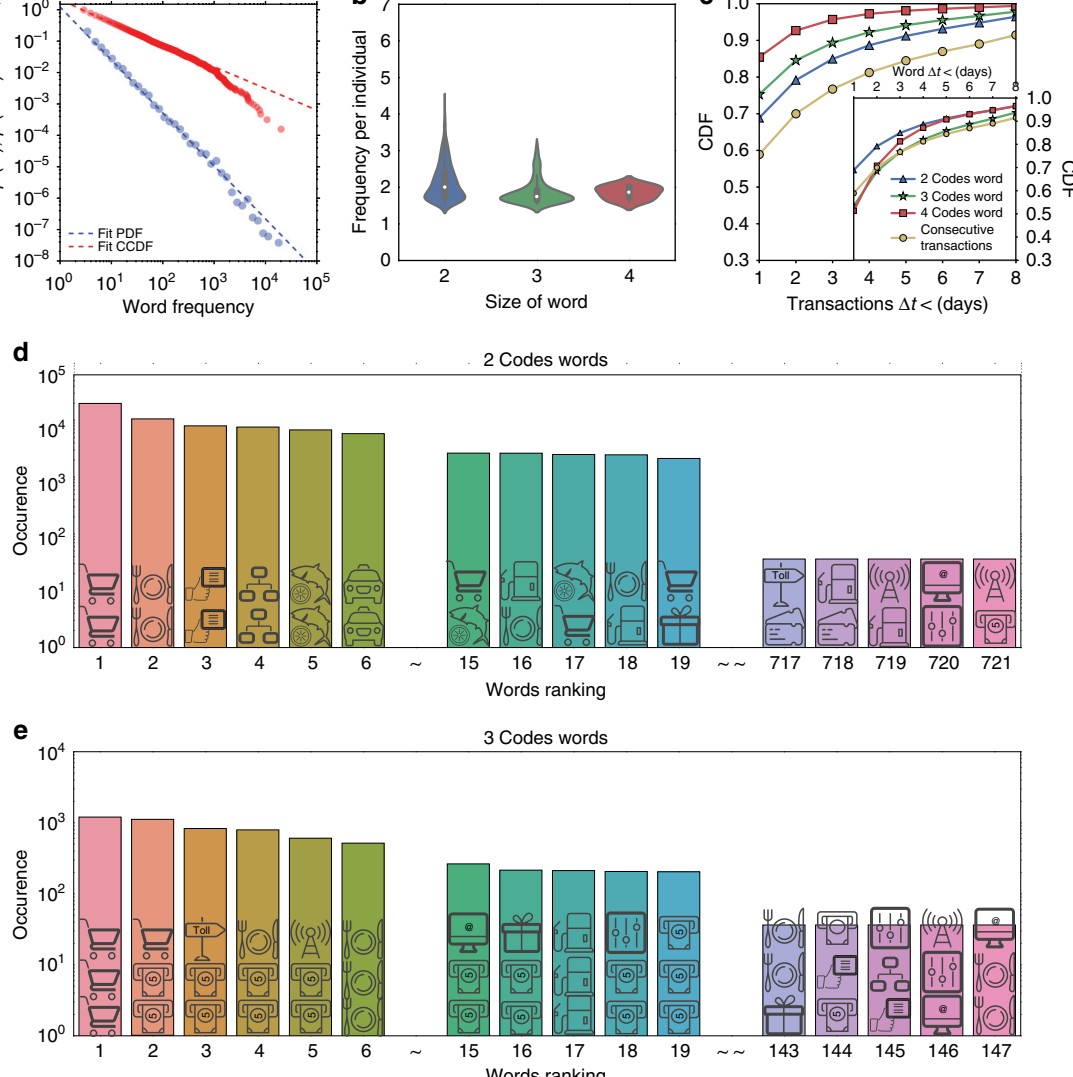

**Fig. 3** Semantic analysis of transaction sequences. **a** Probability density function plot of the occurrence of words $\{w_i\}$ and its complementary cumulative distribution; the probability distribution words manifest a power-law behavior $p(w_i) \propto x_i^{(-1.70)}$, with $x_i$ frequency of the $\{w_i\}$ and Kolmogorov–Smirnov distance $D_n = 0.014$. **b** Distribution of the occurrence of words in the transaction sequences by the word length. **c** Inter-time transactions between purchases. The purchases within each word are more likely to occur within a day with respect to two random consecutive transactions. **c** (inside) Moreover, the purchase time to accomplish a word completely is less with respect to two random consecutive transactions. **d**, **e** Examples of words composed by two and three codes, respectively, ordered by the number of occurrences. The icons used in this figure are work of Azaze11o/Shutterstock. com

word purchases are smaller with respect to two random consecutive transactions. Moreover, the time to perform an *n*-transaction word, defined as the time between the first and the last purchase of the word, is smaller than the time of two consecutive transactions picked randomly (Fig. 3c). The set of words $\{w_i\}$ for user $i$ are significant only if their occurrence differs from the outcome of a random process with the same number of transactions per type. To detect the words that are significant, we generate 1000 randomized code sequences for each user. For each realization, we apply the Sequitur algorithm to define the words in the randomized sequences and evaluate the significance level of the user's words by computing the *z*-score of the occurrence of the real words with respect to the randomized ones. *Z*-score test needs to be performed on a Gaussian distribution of word occurrence. The word-occurrence distribution of simulated samples has in general a normal shape. But in several cases, the frequency of the generated words has a small number of

occurrences; in Supplementary Figures 3, 4, we show the robustness of a *z*-score benchmark to assess the word significance for non-Gaussian distributions. We extract for each user, the set of significant words with *z*-score >2, defined as $\{w_i\}$. The selected words represent the shopping routines that indicate informative choices in the user's spending behavior (see Supplementary Figure 5), given that their occurrence vary from the mean by two standard deviations. In the Supplementary Figure 5D, E, we analyze the number of valid users with at least a significant word depending on the *z*-score threshold.

**The lifestyles.** With these meaningful samples, we can now measure the similarity between shopping behaviors among users. To that end, we decompose each significant word as direct links between its transaction codes. Each user is represented by a directed network, in the space of MCC, that collects all the links present in the user's words. We then calculate the Jaccard

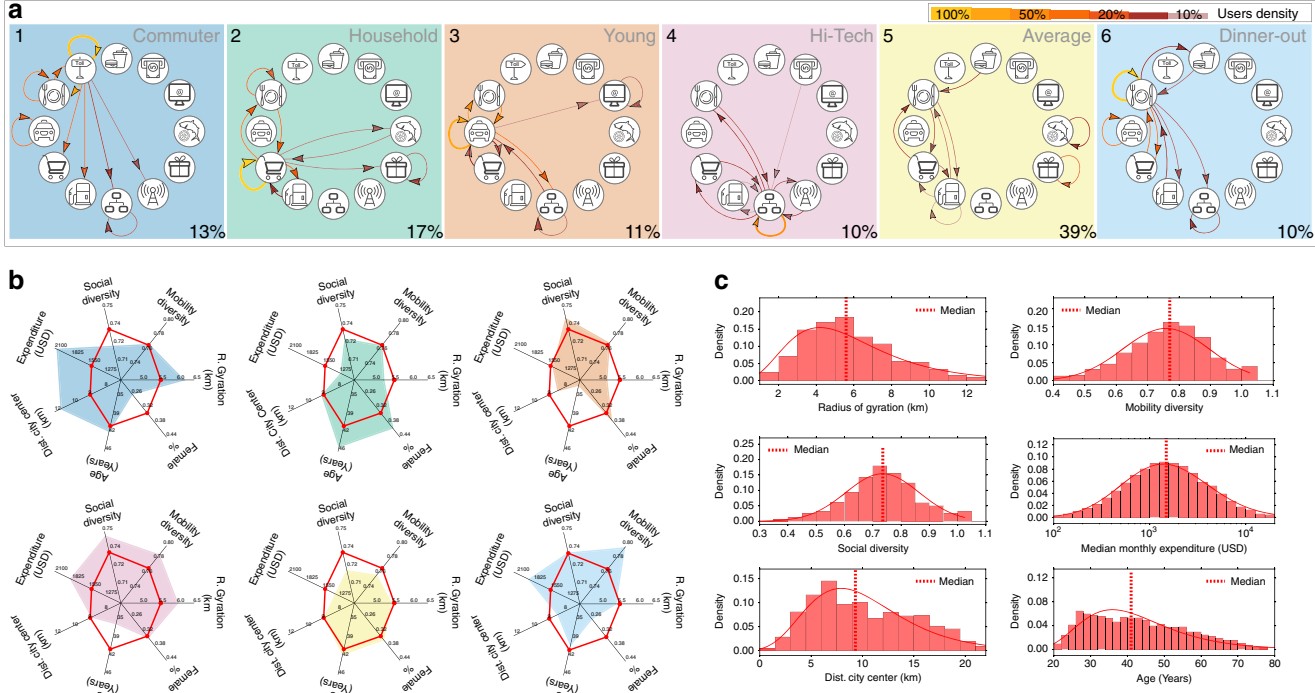

**Fig. 4** Identified lifestyles I. **a** Groups based on their spending habits. We show the top 10 most frequent spending sequences of the users in each group, representing more than 30% of users' shopping routines. The percentage of the total users in each group is shown in the bottom-right corner. **b** Comparison of the median of socio-demographic variables within each group with respect to the median of all users is in red. (The color of the radar plot identifies the spending habits in **a**.) **c** Distribution of individual characteristics among users: gender, radius of gyration, mobility diversity, social diversity, median expenditure by month, average distance traveled from the center of residence zipcode to the city center, and age (see Supplementary Figures 11–16, 21 for further information). The icons used in this figure are work of Azaze11o/Shutterstock.com

similarity coefficient between all the users to compare the set of links in their networks (see the illustration of the method in Fig. 2b). Since user networks have a low degree, our similarity measure is not sensitive to the sets' size (Supplementary Figure 6C). Moreover, our results are in agreement with the ones that use the turnover component of Jaccard dissimilarity index[32], which is less susceptible to the sets' size (see Supplementary Figure 6). Owing to the Jaccard index, we obtain the matrix $M$ of users' similarity in shopping sequences.

Finally, we identify the groups in this matrix by applying a parallel Louvain algorithm for faster unfolding of communities in $M$[33,34]. The same clusters appear with Leading Eigenvector[35] and Walking Trap[36] (Supplementary Figures 7, 8). We detect six clusters or groups of users who share similarities in their spending habits; one of the six encloses unlabeled users who are close to the average behavior, while the other five present interesting behavioral preferences as confirmed later by their demographics and their mobile phone records.

Figure 2c shows the group's shopping habits. The weight of the arrows between two codes represents the fraction of users of a given cluster that have the given transaction sequence. This schematic representation of the group's routines is possible because our method firstly, detects the most significant sequences of transactions and secondly preserves the temporal information embedded in word as the ordered sequence of transaction.

**Coupling credit card data with mobile phone data**. In order to gather a more comprehensive portrait of the users' behavior, we couple the information of the CCR users with their CDR data (Fig. 2d, e). From the mobile phone data, we analyze the basic characteristics of an individual's social contacts and their

mobility network with well-established metrics, namely, social diversity, homophily, mobility diversity, radius of gyration[8,37], tower residual activity[38], and mobility behavioral pattern. Social network diversity is the entropy associated with the number of individual $i$'s communication events with their reciprocal contacts divided by the number of contacts[1]. Homophily, in the call graph from the mobile phone data, is a metric that investigates whether or not two users in the same cluster have a higher probability of contacting each other. Mobility diversity is measured via entropy in the number of trips between locations normalized by the number of visited locations[37]. Ego networks are defined by a focal node (ego) and the users to whom the ego is directly connected. High diversity score in the ego network implies that individuals split three times evenly among their social ties. High diversity in the network of trips among locations means that individuals distribute their number of trips evenly among their visited urban locations. Radius of gyration, in turn, defines the radius of the circle within which they are more likely to be found, it is centered in all the visited locations of $i$ and weighted by the number of mobile phone records in each location[8]. From the urban science perspective, we investigate the cell towers' residual activity as defined by Toole et al.[38] to determine whether users who belong to the same cluster tend to aggregate in a specific area of the city. Residual activity can be interpreted as the amount of mobile phone activity in a region relative to the expected mobile phone activity in the whole city. Finally, to assess the mobility behavioral pattern, we analyze the portion of explorers and returners among the users[39]. Returners are the users who limit much of their mobility to a few locations; in contrast, the explorers have a tendency to wander between a larger number of different locations.

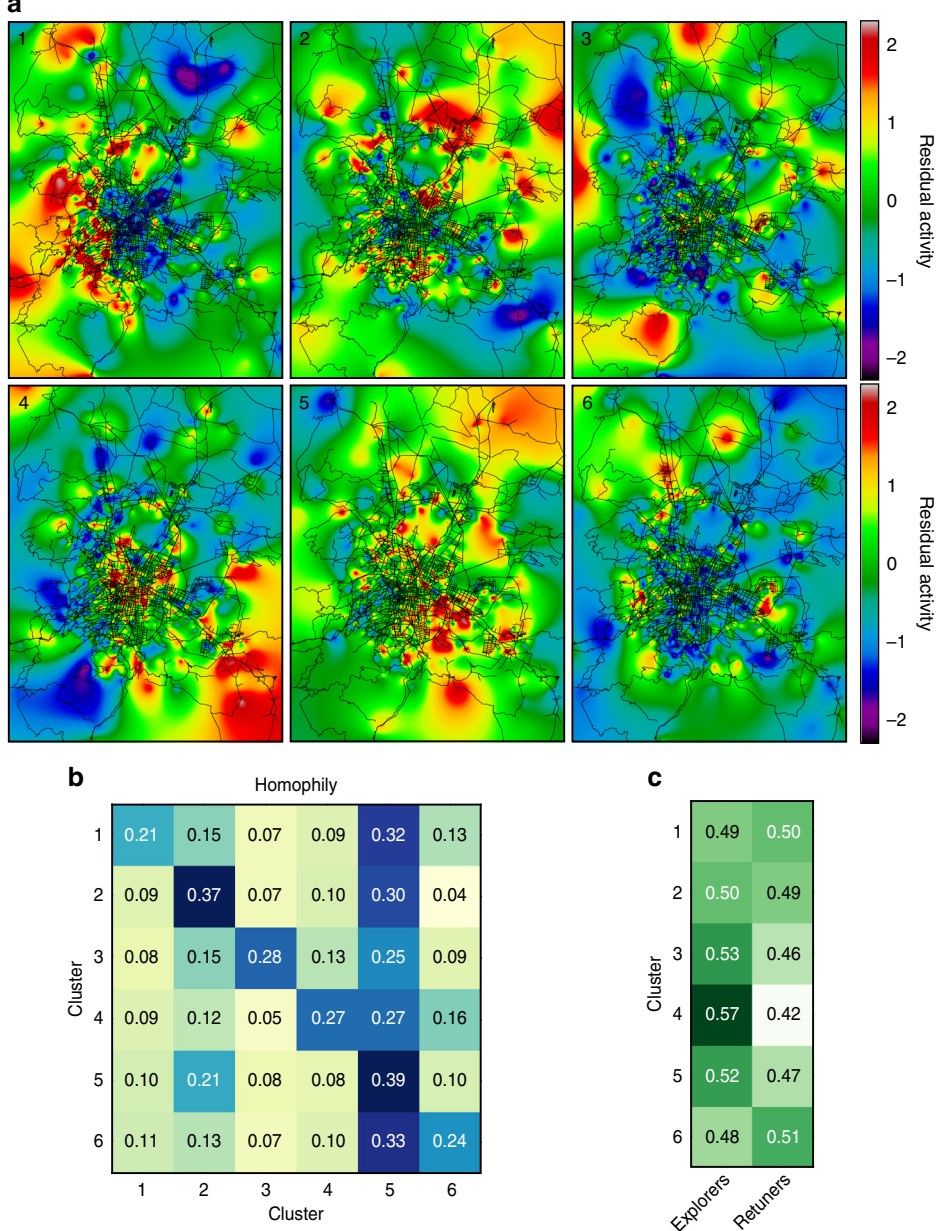

**Fig. 5** Identified lifestyles II. **a** Cell towers residual activity by clusters. **b** Clusters' homophily. As expected, each user tends to contact the users that belong to the same clusters or cluster 5 "uncategorized," which is the cluster with the highest number of users. Remarkably, there is a slight preference to contact cluster 2, the homemakers, which represent the oldest group. **c** Distribution of returners and explorers across the clusters (see Supplementary Figure 11–16, 21 for further information). Maps in this figure were created using the software QGIS using OpenStreetMap data

## Discussion

Five of the six clusters detected depict a particular lifestyle on how individuals spend their money, move, and contact other individuals. One transaction type is at the core of the spending activities in each group, and 90% of the users within the cluster have it repeated as a sequence (or significant word, represented by yellow loop in Fig. 4a). This transaction also appears in more than 45% as starting or ending transaction of the sequences of other types of transactions within the group (Fig. 4a). The users clustered by using our approach have relatively high Shannon entropy in their transactions and a Sequitur compression ratio of 1.5 or larger (Supplementary Figure 10). Cluster 5 aggregates the uncategorized users. In particular, users who belong to this cluster have less than five significant sequences and less variation in their expenditure types (Supplementary Figures 7–9).

Figures 4b, 5 show that each cluster reveals consistent relations between expenditure patterns and age, mobility, and social networks of their members, hinting that the method actually unravels behavioral groups in the data or actual lifestyles. Cluster 1 aggregates users whose core transaction is toll fees, and accordingly we label them as Commuters. They live furthest from the city center, expend the most, travel longest distances, and are majority male, as confirmed from the analysis of the radius of gyration and the residual activity in Fig. 5a. Conversely, users in the cluster 2 or homemakers have grocery stores as a core transaction. They represent the oldest group with least expenditure, mobility, and a larger share of women. Although the social network of this cluster manifests a lower diversity, there is a slight preference in the homophily matrix in this cluster, suggesting that the few connections are cluster transversal (Fig. 5b). Younger

users are split into two groups (clusters 3 and 4) with different values in their expenditure, and social and mobility diversity. Cluster 3 is labeled as Youths because it has the youngest individuals with taxis as their core transaction. Cluster 4 is close in age to cluster 3, but has computer networks and information services as a core transaction. They are labeled as Tech users and have higher than average expenditure and higher diversity in their social contacts and mobility networks. The residual activity (Fig. 5a) suggests that their movements are within the city center. Moreover, clusters 3 and 4 are the only ones with a majority of explorers within their users, supporting the lifestyle fingerprint (Fig. 5c). Finally, cluster 6, labeled as Diners, aggregates middle-aged users who have restaurants as their core transaction with high mobility diversity and higher expenditures (see Supplementary Figures 11–16, 21 for further information).

We compare the detected groups with the ones extracted via the patients' stratification technique to analyze the health records[24]. Instead of applying the Sequitur algorithm to assess the likelihood of a given sequence of codes, we compute, for each user's code, the TF-IDF frequency measure[40], which rewards high code frequency in the individual records and penalizes high prevalence across the all user's history. The similarity matrix among users is based on the cosine similarity in the space of the code frequency TF-IDF. The clusters extracted via this method (Supplementary Figure 17) do not have socio-demographic similarities, and the characteristics of the members within each group average similarly to the population. Moreover, TF-IDF does not disentangle the Zipf distribution (Supplementary Figure 17c), meaning each cluster keeps the same overall transaction frequency.

Furthermore, we compare our clusters with the LDA[27,41]. This method first identifies five topics represented by an ensemble of MCCs. Each user is identified by a vector $v_i$ weighting the mixture of those five topics. We compute the users' similarity matrix using Jensen–Shannon divergence[42] among $v_i$. Finally, we perform the Louvain algorithm over the matrix. Four of the seven identified clusters (1, 2, 3, and 7), in the Supplementary Figure 18, are similar to our clusters (1, 2, 3, and 6). Furthermore, the LDA is able to untangle the similar variance from the Zipf distribution (Supplementary Figure 18C) compared with our method (Supplementary Figure 13B).

With respect to the above-mentioned methods (TD-IDF and LDA), our approach deconstructs the Zipf distribution into the constituents' behavior (see Supplementary Figure 13B). The resulting clusters of the latter are comparable with our method. Furthermore, our framework is able to capture the routines of each cluster as ordered sequence of transaction; this temporal information is lost using the above-mentioned approaches. These tests stress the effectiveness of our method.

Finally, we apply our framework to another minor city of Mexico: Puebla (Supplementary Figure 19–21). As already shown by Sobolevsky et al.[23], different cites manifest a general behavior in terms of spending patterns, maintaining some unique characteristics. In Puebla, we detect six clusters; four of them share similar routines and attributes to the main city (Mexico City clusters (2, 3, 5, and 6)). Comparing the median absolute deviation of each cluster, it is possible to assess the diversity of every socio-demographic attribute (Supplementary Figure 21). In particular, the routines of Commuters' clusters are identifiable in both of the cities, with some difference in the mobility attributes. Finally, in Puebla, the Youth cluster is replaced with one with different core transactions in the miscellaneous food store and insurance instead of taxi and restaurants. This result stresses how our framework can capture cities' differences in terms of spending patterns, providing a tool to enrich the urban activity models.

Taken together, we present a method to detect behavioral groups in chronologically labeled data. It could be applied also to similar data sets with Zipf-like distributions, such as disease codes in patients' visits[24,25] or law-breaking codes in police databases[43]. Given the ubiquitous nature of the CCR transaction distribution by type[23], similar groups could be detected and compared among cities worldwide. Analogous to the price index that uses online information to improve survey-based approaches to measure inflation[44], the meaningful information of groups extracted from the CCR data can be used to compare consumers worldwide[4]. Interesting avenues for the application of this method are policy evaluation of macroeconomic events such as inflation and employment and their effects on the spending habits of various groups[45].

## Methods

**Credit card data sets**. Credit card data sets, also referred to as CCRs, used in this study consists of 10 weeks of records, starting from the 1st week of May 2015, of all the credit card users of a particular bank across each subject city. Each individual CCR consists of a hashed user identification string, the time stamp of the transaction, the associated expenditure labeled with the transaction type via an MCC[29], and the transaction amount. For each user, the data set contains age, gender, and residential zipcode of the user (Supplementary Figure 1A–C).The purchase entries are aggregated by user and are temporally ordered with respect to each day.

**Mobile phone data sets**. Mobile phone data sets, also referred to as CDRs, used in this study consist of 6 months of records, starting form March 2015, of all mobile phone users of a particular carrier across each subject city. Each individual CDR consists of a hashed user identification string, a time stamp, and location of the activity. The spatial granularity of the data varies between cell tower levels.

**Census data**. The census data used in this work were download from the Instituto Nacional de Estadística Geografía e Informática, México (http://www.inegi.org.mx/ last checked 13/Jun/2018). In particular, the data regarding the population distribution among the districts are from "Source: INEGI, Intercensal Survey 2015" and the data on the district income are from "Source: INEGI, National Survey of Occupation and Employment (ENOE). Population aged 15 years and older."

**Data availability**. For contractual and privacy reasons, the raw data is not available. Upon request, the authors can provide the data of the matrix of user similarity along with appropriate documentation for replication.

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

## Acknowledgements
This work was supported by the Gates Foundation (grant OPP1141325) and United Nations Foundation (grant UNF-15-738). We acknowledge Rebecca Furst-Nichols and Jake Kendall for planning the study. We also thank Edward Barbour, Philip Chodrow, and Balazs Lengyel for the helpful discussions. Views and conclusions in this document are those of the authors and should not be interpreted as representing the policies, either expressed or implied, of the sponsors. Riccardo Di Clemente as Newton International Fellow of the Royal Society acknowledges support from the Royal Society, the British Academy, and the Academy of Medical Sciences (Newton International Fellowship, NF170505). The icons used in this paper are work of Azaze11o/Shutterstock.com.

## Author contributions
R.D.C. analyzed the data, performed the research, and created the maps, S.X. developed and tested the machine learning algorithm; R.D.C., M.T., M.L.-O., B.V., and M.C.G. planned the study; R.D.C. and M.C.G. designed the study and wrote the paper; and M.C.G. coordinated the study. All authors gave their final approval for publication.

## Additional information

**Competing interests:** The authors declare no competing interests.

