## [Peer Review File · Nature Communications]

Reviewers' comments:

Reviewer #1 (Remarks to the Author):

This manuscript proposes a new framework to reveal life styles in urban populations by analyzing sequences of purchases in credit card data. Since it is apparently not possible to capture differences among bank customers based only on the frequency of transactions per category, the authors favored a different approach focusing on the similarity between customer's shopping mobility networks. The considered network is a directed and unweighted network, where a link between two transaction categories represents a significant occurrence of the two successive transaction categories in the user's sequence of purchases. Recurrent occurrences of successive transactions in the user's sequence of purchases are identified with the Sequitur algorithm and aggregated under the form of "words" (succession of transaction categories). A null model is then proposed to detect a set of significant "words" for each user. Finally, a community detection algorithm is applied on the similarity matrix among users' set of significant "words" to detect clusters of customers exhibiting similar shopping habits.

My overall impression about the paper is positive. The paper is well-written and presents an interesting piece of work. However, as it currently stands, the manuscript contains several shortcomings (listed below) that render it inappropriate for a high impact journal like Nature Communications.

1) Zipf distribution

It is not clear to me where the authors found in references 23 and 30 that the "Zipf-like behavior has been observed in various cities" and that "it manifests the same features if we group consumers depending on socio-demographics attributes". Actually in their conclusion the authors of reference 30 claim that "Exploring the connection between individual economic activity and demographics, we extended the findings of [28], showing that age and gender have a major impact on all of the bank card spending parameters." Note that the reference [28] mentioned by the authors (which correspond to the reference (13) in the manuscript) shows differences according to the gender and age in the fraction of money spent by business category. I recommend the authors to clarify this point in the introduction.

2) Data

- I am aware that datasets containing individual information provided by private companies are complicated and sometimes impossible to share, even upon request, but at least the authors should disclose the name of the cities studied in the paper.
- If census data are accessible, it would be interesting to evaluate the representativeness of the sample by comparing the proportion of users according to the gender and the age in the credit card database with national or local statistics.
- The study focuses on the analysis of the individual sequences of purchases and spending habits but there is no information about the number of transaction per day. I would strongly encourage the authors to add a figure displaying the distribution of transactions per day according to the customers' socio-demographic characteristics.

3) Methods

- My main concern is that there is no limit of time between two consecutive transactions, only the sequential aspect is considered not the temporal one. In other words, what is the meaning of two

successive events if the inter-event time between them is two days? I would recommend the authors to assess the impact of the maximal inter-event time considered on the results. They could, for example, consider only successive transactions occurring the same day.

- It is not clear to me how the similarity among customers is computed. Is the Jaccard index applied on the network or the set of words? The network is composed of links between two transaction categories while it seems that a word can be composed by more than two categories.
- Moreover, the Jaccard index is quite sensitive to the number of words. I would recommend the authors to only consider the turnover component of Jaccard dissimilarity index (Baselga, 2012).

<http://webspersoais.usc.es/export9/sites/persoais/persoais/andres.baselga/pdfs/Baselga2012.pdf>

4) Group interpretation

I really enjoyed reading this part of the manuscript. It is interesting and relatively rare to see how different (big) data sources can be combined to extract meaningful patterns. Nevertheless, I would recommend the authors to use Z-scores or "test-values" to assess the relative importance of the socio-demographic variables (compare to the average) in the different groups.

Reviewer #2 (Remarks to the Author):

In this paper, the authors cluster credit card users into six clusters, labeling five of them based on user characteristics like expenditure, age, gender as well as information extracted by mobile phone data. The clustering process is based on the application of a text compression technique (the Sequitur algorithm) on the sequence of credit card purchases, which allow the authors to overcome the main shortcoming of existing techniques like TF-IDF and LDA (i.e., not using temporal sequence).

The paper is interesting and clear. In particular, the usage of both credit card data (CCR) and mobile phone data (CDR) can potentially reveal interesting patterns about human behavior. Unfortunately, the potential of the link between the two data types is not fully exploited. Although radius of gyration, social diversity and mobility diversity are interesting measures to describe an individual's social behavior, they provide a very aggregated view on individual human behavior, hiding many interesting patterns that can be extracted with a deeper investigation of CDR.

For example, the role of the social network could be investigated. The authors could reconstruct the call graph from mobile phone data and investigate whether or not two users in the same cluster have a higher probability of being friends in the call graph. This would provide interesting information about the existence of a social homophily in the clusters. As a further example, given a user in a cluster, how do her friends distribute in the six clusters?

From a human mobility perspective, mobile phone data could be exploited in many ways. For example, the authors could investigate the mobility motifs in the six clusters as specified in Schneider et al. (<http://bit.ly/2xfdCQd>) or the proportion of the so-called explorers and returners as defined by Pappalardo et al. (<http://go.nature.com/2pFgqDf>). From an urban science perspective, it could be interesting to investigate whether the users in a cluster tend to aggregate in a specific area of the city.

In summary, profiling credit card users according to their purchase habits is potentially interesting. However, the analysis is based on very aggregated measures and so it is not clear why it is important and what are the consequences of the findings. A deeper investigation of the clusters

based on the CDR data is needed to provide meaningful behavioral patterns, to show why the observed clusters matter in terms of social network behavior and human mobility behavior of the considered individuals.

Detailed comments on the paper:

- in Results: the correlation between the CCRs expenditure  the correlation between the MEDIAN CCRs expenditure

- the authors extract the set of significant words with z-score greater than 2. Why do you choose 2? This is not motivated in the paper. How the results change (in terms of clusters) as this threshold changes should be also investigated.

- Discussion: the meaning of the values of radius, social and mobility diversity should be better explained for readers that are not expert in human mobility analysis. For example, in cluster 1 the high radius of gyration well supports the labeling proposed by the authors, i.e., Commuters. I suggest to better explain the contribution of social and mobility measures in defining the cluster labeling.

- Discussion: the authors use the Louvain algorithm with a threshold of 0.1 in the matrix. Why do you choose 0.1? This choice is not justified in the paper.

We are glad to submit the revised version hoping that it is now suitable for publication. Here we discuss in details all the referee points (in red our responds).

Reviewer #1 (Remarks to the Author):

This manuscript proposes a new framework to reveal life styles in urban populations by analyzing sequences of purchases in credit card data. Since it is apparently not possible to capture differences among bank customers based only on the frequency of transactions per category, the authors favored a different approach focusing on the similarity between customer's shopping mobility networks. The considered network is a directed and unweighted network, where a link between two transaction categories represents a significant occurrence of the two successive transaction categories in the user's sequence of purchases. Recurrent occurrences of successive transactions in the user's sequence of purchases are identified with the Sequitur algorithm and aggregated under the form of "words" (succession of transaction categories). A null model is then proposed to detect a set of significant "words" for each user. Finally, a community detection algorithm is applied

on the similarity matrix among users' set of significant "words" to detect clusters of customers exhibiting similar shopping habits.

My overall impression about the paper is positive. The paper is well-written and presents an interesting piece of work. However, as it currently stands, the manuscript contains several shortcomings (listed below) that render it inappropriate for a high impact journal like Nature Communications.

We thank the reviewer for the careful review of our paper. We appreciate the constructive comments on our manuscript. Moreover, we greatly appreciate his/her interest and his/her positive assessment of our manuscript. We considered each comment as detailed below.

1) Zipf distribution

It is not clear to me where the authors found in references 23 and 30 that the "Zipf-like behavior has been observed in various cities" and that "it manifests the same features if we group consumers depending on socio-demographics attributes". Actually in their conclusion the authors of reference 30 claim that "Exploring the connection between individual economic activity and demographics, we extended the findings of [28], showing that age and gender have a major impact on all of the bank card spending parameters." Note that the reference [28] mentioned by the authors (which correspond to the reference (13) in the manuscript) shows differences according to the gender and age in the fraction of money spent by business category. I recommend the authors to clarify this point in the introduction.

1) Zipf distribution-Answer

We thank the reviewer for the observation. We corrected the reference 22 and 23. In particular, we removed the reference 23 and changed the reference 22 (now: W. Matheny et al., The State of Cash: Preliminary Findings from the 2015 Diary of Consumer Payment Choice FedNote, (November 2016)). The reference 22 offers a survey that shows the Zipf-like behaviors of the frequencies of purchases. Moreover, we followed the suggestion of the reviewer pointing out the relations between the Zipf and the social-demographic attributes in the introduction. We highlighted the relation between expenditure and age, gender and socio-economic features in the texts. We moved the ref. 28 of the previous version of the paper, to ref. 23 in order to support these findings. It is important to note that the differences in the abundance of the most frequent purchases category are not substantial between gender and age. While the order in abundance of the less frequent purchases category changes with the social-demographic features. Moreover, the detected differences in life styles are more distinctive signature of grouped individuals.

2) Data

- a) *I am aware that datasets containing individual information provided by private companies are complicated and sometimes impossible to share, even upon request, but at least the authors should disclose the name of the cities studied in the paper.*
- b) *If census data are accessible, it would be interesting to evaluate the representativeness of the sample by comparing the proportion of users according to the gender and the age in the credit card database with national or local statistics.*
- c) *The study focuses on the analysis of the individual sequences of purchases and spending habits but there is no information about the number of transaction per day. I would strongly encourage the authors to add a figure displaying the distribution of transactions per day according to the customers' socio-demographic characteristics.*

2) Data-Answer

- a) We added the references of the cities analyzed in the text. The main city is Mexico City and the second analyzed city is Puebla. Moreover, we added ref. 30 in the main paper in relation of the credit card adoption in Mexico.
- b) We modify Fig.1A in the main text to add the information regarding the 16 districts of Mexico City. We added figure S1d-S1e-S1f to describe the CCR population in each district in comparison with the census data from INEGI disentangling the age and gender relations. We added a sentence in the main text to outline these results.

New Fig.1A (main text): We added the color and the reference to each district. In this way the readers can match the information retrieves by the census in the figures S1b-S1d-S1e-S1f

New Figures S1b-S1d-S1e-S1f (sup info.): We proposed a descriptive study of the credit card users respect the population of each districts of Mexico City. S1b: District Population Vs. Users Districts. S1d: Age distribution of user population vs census population for each district. S1e: Districts names. S1f: Percentage female users vs. districts.

c) We thank the reviewer for the interesting comment. We carefully checked the number of transactions per day for each user depending by the socio-economic characteristics. There is no difference according the consumers' socio

demographic characteristic. We added two figures in the supplementary (see Fig. S2b-c) to tackle this question and we added a sentence in the main text.

New Fig S2b-S2c: (B) Median's distributions of the number of transactions per day divided by socio-demographic features. (C) Percentage of the number of transactions per day per user divided by socio-demographic features.

3) Methods

- My main concern is that there is no limit of time between two consecutive transactions, only the sequential aspect is considered not the temporal one. In other words, what is the meaning of two successive events if the inter-event time between them is two days? I would recommend the authors to assess the impact of the maximal inter-event time considered on the results. They could, for example, consider only successive transactions occurring the same day.*
- It is not clear to me how the similarity among customers is computed. Is the Jaccard index applied on the network or the set of words? The network is composed of links between two transaction categories while it seems that a word can be composed by more than two categories.*
- Moreover, the Jaccard index is quite sensitive to the number of words. I would recommend the authors to only consider the turnover component of Jaccard dissimilarity index (Baselga, 2012).*

3) Methods-Answer

- We appreciate the comment of the referee. Our data set only contains the date of the purchase. We do not have any reference to the hour or minutes of the day when the purchases occurred. However, we know the temporal order of the purchases within each day. Following the suggestion of the reviewer we further disentangled the inter-event-time relation between the transaction within each word and the transactions of a random sequence of consecutive purchases. We observed that in word purchases are indeed more likely to occur within a day*

respect the same number of random consecutive transactions (Fig. 3c in the main paper). Moreover, we analyzed the inter-event-time between the first and the last transaction of each word. We can see that the purchase-time to perform a word is less than two random consecutive transactions. To answer this concern, we improved the description of the data in the main text, we added an analysis on the inter event time between transactions presented in Fig 3c.

New Fig. 3c. Inter-time transactions between purchases. The purchases within each word are more likely to occur within day respect two random consecutive transactions. (C inside) Moreover, the purchase-time to accomplish a word completely is less respect two random consecutive transactions.

- b) Thanks for the comment, we have added the clarification in the text. In particular: we decompose each significant word as directed links between transaction codes. Each user is represented by a directed network, in the space of MCC, that collects all the links present in the user's words. We then calculate the Jaccard similarity coefficient between all the users to compare the set of links in their networks. We improved the main text to be more effective and understandable.
- c) We are thankful for pointing this alternative as valuable approach to assess the comparison between sets. Since the users' networks have a low degree our similarity measure is not sensitive to the sets' size (see Fig. S6c). Moreover, in our case, either the Jaccard index and the turnover of Jaccard dissimilarity detect 6 clusters with similar characteristics (see new Fig. S6a). We added, in the main text, a sentence and the reference (ref.[32]) to the work of (Baselga 2012) to let the reader aware of the size sets' size sensitiveness if he/she will use our method in a different contest.

New Fig. S6 (A) Clustering results using the turnover component of the Jaccard dissimilarity index [32]. (B) Distributions comparison between the Jaccard Similarity and the Jaccard turnover. (C) CDF of the number of links per users' network.

4) Group interpretation

I really enjoyed reading this part of the manuscript. It is interesting and relatively rare to see how different (big) data sources can be combined to extract meaningful patterns. Nevertheless, I would recommend the authors to use Z-scores or "test-values" to assess the relative importance of the socio-demographic variables (compare to the average) in the different groups.

4) Group Interpretation-Answer

Taking in consideration the comment of the reviewer we updated the Fig. S21 using the Median Absolute deviation (MAD). Given that each of the socio-demographic attributes has a log-normal distribution and in the main paper we always analyzed the median value of each attributes, we believe that the MAD is a suitable tool to outline the relative importance of these variables. Moreover, in Fig. S21 shows a comparison between the two cities analyzed (Puebla and Mexico City). We kept Fig 4 of the main paper with the real values, since we believe that this version could convey easily our message to a wider audience.

Fig S21 Analysis of median socio-demographic-mobility index variation per cluster. The y axis represents $\frac{(\bar{x}_i - X)}{MAD(X)}$; with X the median of the whole dataset for the socio-demographic-mobility attribute X , \bar{x}_i the median of the same socio-demographic-mobility attribute of the i -cluster users and MAD the Median Absolute Deviation. Remarkably the behaviors of the clusters (2,4,5,6) are very similar between the two cities considered. The two Clusters 3 as already stress represent two different segments of the population. Meanwhile the clusters 1 of the commuters have different behaviors maintaining the lower transaction diversity this could be due to the different topology of the cities.

Reviewer #2 (Remarks to the Author):

In this paper, the authors cluster credit card users into six clusters, labeling five of them based on user characteristics like expenditure, age, gender as well as information extracted by mobile phone data. The clustering process is based on the application of a text compression technique (the Sequitur algorithm) on the sequence of credit card purchases, which allow the authors to overcome the main shortcoming of existing techniques like TF-IDF and LDA (i.e., not using temporal sequence).

The paper is interesting and clear. In particular, the usage of both credit card data (CCR) and mobile phone data (CDR) can potentially reveal interesting patterns about human behavior. Unfortunately, the potential of the link between the two data types is not fully exploited. Although radius of gyration, social diversity and mobility diversity are interesting measures to describe an individual's social behavior, they provide a very aggregated view on individual human behavior, hiding many interesting patterns that can be extracted with a deeper investigation of CDR.

We thank the reviewer for the careful review of our paper. We appreciate the constructive comments on our manuscript. Moreover, we greatly appreciate his/her interest and his/her positive assessment of our manuscript. We followed carefully the recommendations

- 1. For example, the role of the social network could be investigated. The authors could reconstruct the call graph from mobile phone data and investigate whether or not two users in the same cluster have a higher probability of being friends in the call graph. This would provide interesting information about the existence of a social homophily in the clusters. As a further example, given a user in a cluster, how do her friends distribute in the six clusters?*

Answer

We took in consideration the comment of the reviewer analyzing the Homophily among the clusters. Each user is more likely to contact users that belong to the same cluster or that belong to cluster 5 that aggregates the uncategorized users. Remarkably there is a slightly higher preference to contact the cluster 2 (the *Homemakers*). This is in agreement with the socio-demographic attributes extracted from the cluster 2 that represent the oldest group of users. We added in the main text Fig. 5b and the previous observation in the discussion section.

New Fig. 5b: Clusters' homophily. As expected each user tends to contact the users that belong to the same clusters or the cluster 5 that aggregates the uncategorized users. Remarkably there is a little preference to contact the cluster 2 of the homemakers, among the others

- From a human mobility perspective, mobile phone data could be exploited in many ways. For example, the authors could investigate the mobility motifs in the six clusters as specified in Schneider et al. (<http://bit.ly/2xfdCQd>) or the proportion of the so-called explorers and returners as defined by Pappalardo et al. (<http://go.nature.com/2pFgqDf>). From an urban science perspective, it could be interesting to investigate whether the users in a cluster tend to aggregate in a specific area of the city. In summary, profiling credit card users according to their purchase habits is potentially interesting. However, the analysis is based on very aggregated measures and so it is not clear why it is important and what are the consequences of the findings. A deeper investigation of the clusters based on the CDR data is needed to provide meaningful behavioral patterns, to show why the observed clusters matter in terms of social network behavior and human mobility behavior of the considered individuals.*

Answer

We appreciate these suggestions of the reviewer. We analyzed two distinctive aspects of the users' mobility:

a) An urban approach in which we disentangle the differences by clusters of the towers' residual activity. Defined by the new ref. [38] (J. Toole et al., Proceedings of the ACM SIGKDD international workshop on urban computing, 1-8, (2012)), the residual activity can be interpreted as the amount of mobile phone activity in a region relative to the expected mobile phone activity in the whole city. In this way, we can quantify the city areas where each cluster is more active. Interestingly, each cluster has well-defined areas of mobility as depicted in Fig. 5a.

New Fig. 5A Cell towers residual activity by clusters

b) As suggested by the reviewer we also investigated the findings of Pappalardo et al. within our clusters. In particular, the cluster 4 Tech-Users, show a majority of explorers in their users. This is in agreement with our findings - the Tech-Users have higher mobility and social diversity -. We added Fig. 5c in the main paper and Fig. S14-S16.

(Right) New Fig 5C of the main text Distribution of returners and explorers across the clusters. (Left) New fig. S14. Returners and Explorer analysis all the users.

New Fig. S15 Explorer and returners by clusters

New Fig. S16 Returners and Explorer cell tower residual activity following *

Detailed comments on the paper:

- in Results: the correlation between the CCRs expenditure  the correlation between the MEDIAN CCRs expenditure

Corrected.

- the authors extract the set of significant words with z-score greater than 2. Why do you choose 2? This is not motivated in the paper. How the results change (in terms of clusters) as this threshold changes should be also investigated?

We selected Z-score=2 as threshold since it represents a deviation of 2 sigma from the mean values (approx. 95% of the distribution). Selecting higher values of the z-score generate a decrease of number of users (see new Fig. S5d). While lower values of the z-score will impact the significance of the word in our context (see new Fig. S5e). We believe that using a standard threshold as 2 sigmas is effective in our framework. We added a sentence in the paper to explain our choice and the two figures (Fig. S5d-e) in the supplementary information.

- Discussion: the meaning of the values of radius, social and mobility diversity should be better explained for readers that are not expert in human mobility analysis. For example, in cluster 1 the high radius of gyration well supports the labeling proposed by the authors, i.e., Commuters. I suggest to better explain the contribution of social and mobility measures in defining the cluster labeling.

We followed the suggestion of the reviewer and we added further explanations in the discussion section of the main text.

- Discussion: the authors use the Louvain algorithm with a threshold of 0.1 in the matrix. Why do you choose 0.1? This choice is not justified in the paper.

We thank the reviewer for this observation. The threshold of 0.1 was chosen only to maximize the time efficiency of the clustering algorithms in the TDIF case. We performed our analysis without the threshold. The result was the same as before, we removed the threshold.

REVIEWERS' COMMENTS:

Reviewer #1 (Remarks to the Author):

The authors properly addressed my comments and the manuscript is substantially improved, I don't have any major objections to acceptance.

Reviewer #2 (Remarks to the Author):

With respect to the previous version of the paper, the authors added several analyses. However, the main limits of the paper are the same as the previous version.

Basically, the authors perform a clustering of the users based on a limited set of existing aggregated features, using existing techniques. Hence, neither the methodology nor the proposed aggregated measures are a novel contribution of this paper.

The main discovery is the fact that there are 6 clusters of users, named by the authors according to the typical values of the chosen aggregated measures. It is still not clear why and how this discovery is important. The analysis of the homophily of users within clusters is simplistic and does not reveal interesting results (e.g., the users tend to communicate preferably with cluster 5, but what does it mean? Why it is an interesting fact?)

Also, I would have expected a deeper analysis of the mobility (or social) habits of users, relating for example their typical movements, and the times they perform them, with the type of purchase they make with credit cards. Is the type of purchase (e.g., grocery stores, eating places, etc.) predictive of the next movement (and/or social interaction) of a user? Vice versa, is it the movement (and/or social interaction) of a user predictive of the next purchase type? The big potential of the proposed research is in the combination of CDRs and CCRs to provide a more comprehensive picture of an individual's behavior. Nevertheless, this potential is still not exploited in the paper.

The analysis provided, though interesting, is very preliminary.

Reviewer #2 (Remarks to the Author):

With respect to the previous version of the paper, the authors added several analyses. However, the main limits of the paper are the same as the previous version. Basically, the authors perform a clustering of the users based on a limited set of existing aggregated features, using existing techniques. Hence, neither the methodology nor the proposed aggregated measures are a novel contribution of this paper.

We appreciate the careful review of our paper. As outlined in the reply, we performed all the analysis he suggested. We would like to emphasize that the main aim is a novel method to uncover patterns of collective behavior extracted from the credit card data. Specifically, how the digital footprint of CCRs can be used to detect spending habits, reflecting interpretable lifestyles in the population at large. We combine methods of analysis in a framework that deconstructs Zipf-like distribution into its constituents' distributions, separating behavioral groups. The aggregate measures proposed in the paper are used to measure the attributes of the groups detected.

The main discovery is the fact that there are 6 clusters of users, named by the authors according to the typical values of the chosen aggregated measures. It is still not clear why and how this discovery is important. The analysis of the homophily of users within clusters is simplistic and does not reveal interesting results (e.g., the users tend to communicate preferably with cluster 5, but what does it mean? Why it is an interesting fact?)

Most of the features detected in human groups are intuitive. The interesting aspects is to be able to label passive data, posing tremendous value to their uses for urban applications without the need of surveys. Specifically, our framework provides insights into collective behaviors in unprecedented way. We used the semantic of spending activities to unravel types of consumers.

As suggested by the referee we improved the explanation of the homophily results in figure's caption and in the text. As outline by the referee the users of the other clusters tend to communicate to the cluster 5, that represent the uncategorized users. The cluster 5 aggregate the majority of the users 39%. The attention should be focus in the other clusters; such as cluster 2. While the social network of this cluster manifests a lower diversity, there is a preference in the homophily matrix to this cluster.

The aggregate analysis was made only to post check the socio-economic attributes of the groups detected. We agree that our work can motivate further analysis and studies, to disentangle the correlations between expenditure behaviors and socio-economic attributes, homophily and predictability.

Also, I would have expected a deeper analysis of the mobility (or social) habits of users, relating for example their typical movements, and the times they perform them, with the type of purchase they make with credit cards. Is the type of purchase (e.g., grocery stores, eating places, etc.) predictive of the next movement (and/or social interaction) of a user? Vice versa, is it the movement (and/or social interaction) of a user predictive of the next purchase type? The big potential of the proposed research is in the combination of CDRs and CCRs to provide a more comprehensive picture of an

individual's behavior. Nevertheless, this potential is still not exploited in the paper.

The main focus of our paper is to propose a new technique to deconstruct Zipf-like distribution into its constituents' distributions, separating behavioral groups. We agree with the referee that the big advance of our research is to define a framework that connect movement with spending behaviors.

Building a proper predictive model that relates the users' movements with the type of purchase or vice versa, is out of the scope of this manuscript and remains an open question. However, we believe that our approach represents the first step in this direction. We are exploring this avenue in our next project.